# Immunogenicity of Tetravalent Protein Vaccine SCTV01E-2 against SARS-CoV-2 EG.5 Subvaraint: A Phase 2 Trial

**DOI:** 10.3390/vaccines12020175

**Published:** 2024-02-08

**Authors:** Jihai Tang, Qinghua Xu, Chaoyin Zhu, Kun Xuan, Tao Li, Qingru Li, Xingya Pang, Zhenqiu Zha, Jinwei Li, Liyang Qiao, Haiyang Xu, Gang Wu, Yan Tian, Jun Han, Cuige Gao, Jiang Yi, Gui Qian, Xuxin Tian, Liangzhi Xie

**Affiliations:** 1Anhui Provincial Center for Disease Control and Prevention, Public Health Research Institute of Anhui Province, Hefei 230601, China; tjh@ahcdc.com.cn (J.T.); xqh1126@sina.com (Q.X.); kunxuan0723@163.com (K.X.); litao@ahcdc.com.cn (T.L.); lqr@ahcdc.com.cn (Q.L.); pxy@ahcdc.com.cn (X.P.); zhenqiuzha@126.com (Z.Z.); 2Funan County Center for Disease Control and Prevention, Fuyang 236399, China; fnjkzcy@163.com (C.Z.); jmklyq@163.com (L.Q.); fnjkwg@163.com (G.W.); fnjkty@163.com (Y.T.); 3Fuyang Center for Disease Control and Prevention, Fuyang 236030, China; jinwei6282li@163.com (J.L.); haiyang86xu@163.com (H.X.); 4State Key Laboratory of Infectious, Disease Prevention and Control, National Institute for Viral Disease Control and Prevention, Chinese Center for Disease Control and Prevention, Beijing 102206, China; hanjun_sci@163.com; 5Beijing Engineering Research Center of Protein and Antibody, Sinocelltech Ltd., Beijing 100176, China; cuige_gao@sinocelltech.com (C.G.); jiang_yi@sinocelitech.com (J.Y.); gui_qian@sinocelltech.com (G.Q.); xuxin_tian@sinocelltech.com (X.T.); 6Cell Culture Engineering Center, Chinese Academy of Medical Sciences & Peking Union Medical College, Beijing 100005, China

**Keywords:** safety, immunogenicity, SARS-CoV-2, multivalent vaccine, booster

## Abstract

The Omicron EG.5 lineage of SARS-CoV-2 is currently on a trajectory to become the dominant strain. This phase 2 study aims to evaluate the immunogenicity of SCTV01E-2, a tetravalent protein vaccine, with a specific emphasis on its immunogenicity against Omicron EG.5, comparing it with its progenitor vaccine, SCTV01E (NCT05933512). As of 12 September 2023, 429 participants aged ≥18 years were randomized into the groups SCTV01E (N = 215) and SCTV01E-2 (N = 214). Both vaccines showed increases in neutralizing antibody (nAb) against Omicron EG.5, with a 5.7-fold increase and a 9.0-fold increase in the SCTV01E and SCTV01E-2 groups 14 days post-vaccination, respectively. The predetermined statistical endpoints were achieved, showing that the geometric mean titer (GMT) of nAb and the seroresponse rate (SRR) against Omicron EG.5 were significantly higher in the SCTV01E-2 group than in the SCTV01E group. Additionally, SCTV01E and SCTV01E-2 induced a 5.5-fold and a 5.9-fold increase in nAb against XBB.1, respectively. Reactogenicity was generally mild and transient. No vaccine-related serious adverse events (SAEs), adverse events of special interest (AESIs), or deaths were reported. In summary, SCTV01E-2 elicited robust neutralizing responses against Omicron EG.5 and XBB.1 without raising safety concerns, highlighting its potential as a versatile COVID-19 vaccine against SARS-CoV-2 variants.

## 1. Introduction

The evolutionary trajectory of Severe Acute Respiratory Syndrome Coronavirus 2 (SARS-CoV-2) has been characterized by uncertainty. Since 2020, there has been an acceleration in the diversity and changing prevalence of SARS-CoV-2 variants, primarily attributable to mutations in the spike protein of the virus. Some of these mutations have led to significant changes in the disease profile and the prognosis of COVID-19 [1,2,3]. 

Recently, the Omicron EG.5 lineage of SARS-CoV-2 has shown increased prevalence, growth advantage, and immune escape properties [4,5,6,7,8,9,10]. EG.5 evolved from the Omicron XBB.1 subvariant and carries an additional F456L amino acid mutation in the receptor-binding domain (RBD) of the spike protein [11,12]. By 25 June 2023, the global prevalence of COVID-19 linked to EG.5 reached 7.6%, and it escalated swiftly to 17.4% by 9 August 2023 [6]. In China, EG.5 and its subvariants accounted for 24.7% of COVID-19 cases in June, rising to 45% a month later, as reported by the World Health Organization (WHO) [6]. In the United Kingdom, the UK Health Security Agency estimated that EG.5 and its subvariants constituted 14.6% of infections as August began [6]. 

From August 2023 on, EG.5 was associated with a concerning increase in hospitalizations and mortality due to COVID-19 in the United States (U.S.). According to the Centers for Disease Control and Prevention (CDC) in the United States, as of 19 August 2023, EG.5 was responsible for 20.6% of the total COVID-19 cases [5,6]. Concerns have also arisen about the efficacy of existing vaccines against EG.5, primarily due to a spike protein mutation, F456L [13,14]. Laboratory experiments have shown that this mutation can enhance immune evasion against bivalent mRNA vaccine-induced neutralizing antibodies (nAb) [10,15]. Following a risk assessment by the WHO, EG.5 and its sublineages, which include EG.5.1, EG.5.1.1, and EG.5.2, were designated as Variants of Interest (VOIs) on 8 August 2023. Furthermore, the WHO and the Technical Advisory Group on SARS-CoV-2 Evolution (TAGVE) recommend that Member States continue to share information on the growth advantage of EG.5. They also suggest providing sequence information on a weekly or monthly basis, conducting neutralization assays, and assessing the impact of variants such as EG.5 on the effectiveness of COVID-19 vaccines [12].

SCTV01E is a tetravalent COVID-19 protein vaccine composed of the trimeric spike extracellular domain (S-ECD) from four SARS-CoV-2 variants, specifically Alpha, Beta, Delta, and Omicron BA.1, with a squalene-based oil-in-water adjuvant SCT-VA02B. The four antigens were produced with stable CHO cell lines, and subsequent purification involved multiple chromatographic steps to achieve high purities. The purified antigens were then combined with the adjuvant and formulated in a single vial [16]. Preclinical studies in naïve or previously primed C57BL/6J mice showed that SCTV01E exhibited favorable immunogenic characteristics to induce balanced and broad-spectrum neutralizing responses against Omicron sublineages (BA.1, BA.1.1, BA.2, BA.3, and BA.4/5) [16]. Notably, in an immunogenicity trial (NCT 05323461), SCTV01E was shown to induce higher levels of nAb against Omicron BA.1 and BA.5 variants compared to the inactivated vaccine and BNT162b2 by day 28 [17,18]. Subsequently, a phase 3 trial which enrolled 9196 participants from 26 December 2022 to 15 January 2023 demonstrated that SCTV01E displayed efficacies of 79.7% in preventing symptomatic infections and 82.4% in preventing all infections caused by SARS-CoV-2, 14 days post-vaccination (data submitted).

On 22 March 2023, SCTV01E received Emergency Use Authorization from the National Health Commission of the People’s Republic of China, allowing its use as a booster dose for COVID-19 vaccination. SCTV01E-2, an updated version, is produced using the same manufacturing process as SCTV01E. However, SCTV01E-2 incorporates a revised antigen formula that includes the S-ECD of the Beta and Omicron BA.1, BQ.1.1, and XBB.1 variants. 

Here, we present findings from a randomized phase 2 trial assessing the immunogenicity of SCTV01E and SCTV01E-2. This study aimed to evaluate nAb responses against EG.5 and XBB.1 in individuals who had previously completed the primary/booster series of COVID-19 vaccinations. 

## 2. Materials and Methods

### 2.1. Study Design and Participants

This ongoing randomized study comprised two parts: Part A, which involved participants aged 18 years and older, and Part B, which included participants aged 3–17 years. This report focuses on the results from Part A, which evaluated the safety and immunogenicity of SCTV01E-2 in adults and compared its immunogenicity with its progenitor vaccine, SCTV01E. Participants were recruited from the Anhui Provincial Center for Disease Control and Prevention, China. Eligible participants were adults aged ≥18 years who had previously received the COVID-19 vaccine, with a ≥6-month vaccination interval between the last dose COVID-19 vaccine and the signature of the informed consent form (ICF). Participants were excluded if they had a fever (temperature ≥ 37.3 °C) within 72 h before the study vaccination, a history of SARS-CoV-2 infection within 6 months, or a positive result for the nasal/nasopharyngeal/throat swab nucleic acid test or rapid antigen test during screening. Details related to the inclusion and exclusion criteria are provided in the trial protocol. 

The protocol of this study, the written ICF, and other information related to participants were approved by the clinical research ethics board of the Anhui Provincial Center for Disease Control and Prevention (China), (protocol code: SCTV01E-2-CHN-1; date of approval: 20 July 2023). The study was registered on ClincalTrials.gov (NCT05933512) [19]. This trial was conducted in adherence to the Declaration of Helsinki, Good Clinical Practice (GCP) requirements, and related regulations issued by authorities.

### 2.2. Randomization and Masking

Randomization and drug codes were generated and securely stored by a third party. Subsequently, the randomization codes were transmitted to the blinded statistician through an independent third-party statistician upon the completion of the study.

The randomization plan was developed using SAS software (Version 9.4). An Interactive Network Response System (IWRS) (Clinflash IRT Version 2.6.0) was used to randomize the eligible participants prior to the study. Enrolled participants were stratified by age (18–59 years vs. ≥60 years), prior SARS-CoV-2 infection history (yes vs. no), and time interval since last vaccination or infection relative to study vaccination (6–11 months vs. ≥12 months).

### 2.3. Procedures

Eligible adults were randomly allocated in a 1:1 ratio to receive one dose of either SCTV01E-2 or SCTV01E. Initially, a group of 14 individuals aged 18–59 years served as sentinels and underwent observation. After the Independent Data Monitoring Committee (IDMC) evaluated their 7-day safety profiles, additional participants were enrolled. The study aimed to maintain the proportion of participants aged 60 years and above at no less than 40%.

Both SCTV01E-2 and SCTV01E were administered via intramuscular injections into the deltoid muscle on the outer upper arm. Post-vaccination, participants underwent a minimum of 30 min of on-site monitoring. Solicited adverse events (AEs) within 7 days, unsolicited AEs within 28 days following the study vaccination, and serious adverse events (SAEs) and adverse events of special interest (AESIs) within 365 days were documented. Participants were followed up every month after 28 days post-vaccination.

Blood samples were collected on Days 0 (pre-vaccination), 14, and 180. The Day 0 samples were analyzed for anti-SARS-CoV-2 spike S1+S2 ECD IgG, anti-RBD IgM, and nAb levels. Further analyses were conducted on the samples taken on Days 14 and 180, specifically focusing on nAb levels. The geometric mean titer (GMT) for anti-spike RBD IgM was ascertained using a qualitative/semi-quantity enzyme-linked immunosorbent assay (ELISA) kit from Vazyme (Vazyme Biotech Co., Ltd., Nanjing, China), following the manufacturer’s instructions. The GMT for live virus nAb activity was measured through plaque reduction neutralization tests (PRNT), employing methodologies delineated in prior studies [20,21].

Following vaccination, participants were regularly contacted via phone calls, text messages, emails, visits at the site, or other means of communication to inquire about COVID-19-related symptoms, and the frequency of follow-up was adjusted as per the study’s progression. They were followed up every month to collect AEs. Participants were also encouraged to report any COVID-19 symptoms and AEs at any point during the study. Rapid antigen or nucleic acid tests (nasal/nasopharyngeal/throat swabs) were conducted for individuals displaying symptoms suggestive of COVID-19.

### 2.4. Outcomes

In Part A, the primary endpoint was the 14-day post-vaccination GMT and seroresponse rate (SRR) of nAb against the Omicron EG.5 sublineage. Secondary endpoints included the 14-day post-vaccination GMT and SRR of nAb against the emerging SARS-CoV-2 variant XBB.1 and the 180-day GMT and SRR against Omicron EG.5 and XBB.1. Exploratory endpoints covered the first occurrence of symptomatic SARS-CoV-2 infection of any severity, starting from 7 days post-vaccination. 

Safety endpoints comprised the incidence and severity of solicited AEs of SCTV01E-2 from Day 0 to Day 7, unsolicited AEs from Day 0 to Day 28, and serious AEs and AESIs from Day 0 to Day 365. AE severity was graded according to guidelines for adverse event grading in preventive vaccine clinical trials [22]. 

### 2.5. Statistical Analysis

The statistical analyses were conducted utilizing SAS software (version 9.4), employing both descriptive and pre-specified statistical test methods. The sample size for the immunogenicity assessment was determined based on a superiority design, aiming to demonstrate SCTV01E-2′s superiority over SCTV01E in terms of the Geometric Mean Ratio (GMR) and SRR of nAbs against the current Omicron EG.5 variant. 

The superiority statistical hypotheses for the primary endpoint assumed that the lower boundary of GMR between the SCTV01E-2 and the SCTV01E groups against the current EG.5 variants was greater than 1 (H_0_: GMR ≤ 1, H_1_: GMR > 1), and the lower boundary of the difference in the SRR between the two groups was greater than 0 (H_0_: △_SSR_ ≤ 0%, H_1_: △_SSR_ > 0%).

In detail, the quantitative immunogenicity data in the log-transformed scale were analyzed using an analysis of covariance (ANCOVA) model with the covariates being the intervention group, age (18–59 years vs. ≥60 years), prior SARS-CoV-2 infection history, time interval since last vaccination or infection relative to study vaccination (6–11 months vs. ≥12 months), and baseline values (in the log-transformed scale). The Least-Square Geometric Mean Ratio (LS GMR) with a 95% CI for treatment difference between SCTV01E-2 and SCTV01E was estimated from the ANCOVA model. The SRR with a 95% CI for treatment difference between SCTV01E-2 and SCTV01E was estimated using the stratified Miettinen–Nurminen method. The stratification factors included age (18–59 years vs. ≥60 years), prior SARS-CoV-2 infection history, and time interval since last vaccination or infection relative to study vaccination (6–11 months vs. ≥12 months).

Participants were grouped based on the vaccines received. Safety assessments encompassed those who received the study vaccine (Safety Set, SS). Immunogenicity assessments were conducted in the Immunogenicity Per-Protocol Set (I-PPS), which included individuals with a valid immunogenicity test result both pre- and post-vaccination and those who had tested negative for the anti-spike RBD IgM test at baseline. In this study, if participants had evidence of a SARS-CoV-2 infection, their immunogenicity data subsequent to the infection were excluded from the immunogenicity analysis set. However, their immunogenicity data before the infection were still utilized for analysis and their safety assessments remained unaffected by the infections.

## 3. Results

### 3.1. Demographic and Baseline Characteristics

As of the cutoff date (12 September 2023), 430 participants were enrolled in Part A. Of these, 429 participants were randomly assigned in a 1:1 ratio to receive either SCTV01E-2 (N = 214) or SCTV01E (N = 215), as shown in Figure 1; 258 (60.1%) participants were aged 18–59 years and 199 (46.4%) participants were male. Demographic characteristics were well balanced between the two groups (Table 1). The mean (SD) body mass index (BMI) was 26.5 (3.5) in the SCTV01E-2 group and 26.2 (3.5) in the SCTV01E group. A small percentage of participants had positive anti-spike RBD IgM (5.1% in the SCTV01E-2 group, 7.9% in the SCTV01E group), while a similar proportion of participants had a history of SARS-CoV-2 infection (26.2% in SCTV01E-2, 26.5% in SCTV01E). Pre-existing comorbidities were reported by 45.8% in the SCTV01E-2 group and 39.1% in the SCTV01E group.

For the immunogenicity per-protocol set (I-PPS) analysis, 399 participants were included (203 in SCTV01E-2 and 196 in SCTV01E), and their demographic characteristics were also well balanced (Appendix A).

### 3.2. Geometric Mean Titer (GMT) and Seroconversion Rate (SRR) of Neutralizing Antibodies against Omicron EG.5 

Fourteen days post-vaccination, the GMT of nAb against the live Omicron EG.5 subvariant was 924 (95% CI: 823, 1037) in the SCTV01E-2 group and 510 (95% CI: 454, 573) in the SCTV01E group, with a 9.0- and 5.7-fold change over baseline (Figure 2). The Least-Square Geometric Mean Ratio (LS GMR) of SCTV01E-2 to SCTV01E was 1.8 (95% CI: 1.5, 2.1) (*p* < 0.001), meeting the predetermined criteria for superiority. 

The SRR for nAb against the Omicron EG.5 was 78.9% (157/199) in the SCTV01E-2 group and 61.6% (117/190) in the SCTV01E group, resulting in a differential SRR of 17.3% (95% CI: 8.3%, 26.1%), which also met the predefined criteria for superiority.

The immunogenicity analysis on the full protocol is presented in the Appendix A.

### 3.3. Geometric Mean Titer (GMT) and Seroconversion Rate (SRR) of Neutralizing Antibodies against Omicron XBB.1

Fourteen days post-vaccination, the GMTs of nAb against the live Omicron XBB.1 were 1887 (95% CI: 1686, 2112) in the SCTV01E-2 group and 1435 (95% CI: 1267, 1626) in the SCTV01E group, showing a 5.9- and 5.5-fold increase from baseline (Figure 3). The LS GMR of SCTV01E-2 to SCTV01E was 1.3 (95% CI: 1.1, 1.5) (*p* = 0.005). 

The SRR of nAb against the Omicron XBB.1 was similar between the two groups, with 68.5% (139/203) in the SCTV01E-2 group and 62.4% (121/194) in the SCTV01E group, resulting in a differential SRR of 6.0% (95% CI: −3.3%, 15.3%), (*p* = 0.206).

The immunogenicity analysis on the full protocol is presented in the Appendix A.

### 3.4. Subgroup Analyses of nAb Responses to Omicron EG.5 and XBB.1

Subgroup analyses were conducted by stratifying participants based on sex, age, history of SARS-CoV-2 infection, time intervals between the last dose vaccine/previous infection and study vaccine, whether they had receipt of a booster dose of a COVID-19 vaccine, the type of their last COVID-19 vaccine dose, and the presence of pre-existing comorbidities. 

The results indicated that SCTV01E-2 consistently led to significantly higher GMTs against Omicron EG.5 than SCTV01E in all subgroups (*p* ≤ 0.01). The only exception was observed in participants who had previously received an adenovirus vector vaccine, where similar GMTs were observed between the two groups (*p* = 0.121), as detailed in the Appendix A. 

Similarly, a significantly higher SRR was observed in most subgroups within the SCTV01E-2 group in comparison to the SCTV01E group (*p* ≤ 0.05). However, in participants with a history of prior infection (*p* = 0.203), those who had not completed the booster immunization (*p* = 0.080), those who are male (*p* = 0.091), or those who had previously received an adenovirus vector vaccine (*p* = 0.560), there were no significant differences in SRRs between the two groups (Appendix A).

For Omicron EG.5, SCTV01E-2 elicited a numerically higher increase in GMT (95% CI) among participants aged ≥60 years (1046 [863, 1269]) compared to those aged 18–59 years (850 [736, 981]). The vaccine exhibited similar GMT (95% CI) among participants with pre-existing comorbidities (937 [775, 1134]) and those without pre-existing comorbidities (914 [791, 1057]). Furthermore, the female subgroup demonstrated a more pronounced increase in GMT (95% CI) than the male group (1061 [915, 1230] vs. 787 [658, 940]), as detailed in the Appendix A. 

Regarding Omicron XBB.1, SCTV01E-2 demonstrated numerically higher GMT levels among older participants (≥60 years) in comparison to younger participants (18–59 years) (2101 [1772, 2490] vs. 1754 [1509, 2039]). Similarly, participants with pre-existing comorbidities exhibited numerically higher GMT levels than those without pre-existing comorbidities (1963 [1669, 2308] vs. 1828 [1561, 2141]). (Appendix A).

### 3.5. Vaccine Immunogenicity and Effectiveness Will Continue to Be Monitored 

As of the cutoff date (12 September 2023), the medium (min, max) follow-up was 38 (38, 47) days, and no participant had reported symptomatic SARS-CoV-2 infection. Since the study is still in progress, the long-term decay rate of nAb and the vaccine effectiveness of the study vaccine will continue to be monitored.

### 3.6. Adverse Events

A total of 429 participants were included in the safety set, with 214 in the SCTV01E-2 group and 215 in the SCTV01E group. The overall incidence of AEs was similar between the two groups (SCTV01E-2 vs. SCTV01E, 42.5% vs. 39.1%). Solicited local adverse reactions (ARs) are detailed in Figure 4A and tabulated in the Appendix A, while solicited systemic ARs are summarized in Figure 4B and tabulated in the Appendix A. No treatment-related SAEs, AESIs, or deaths were reported.

The most frequently reported solicited local ARs, with an incidence of 5% or higher, included injection site pain (SCTV01E-2 vs. SCTV01E, 24.3% vs. 24.2%), as shown in the Appendix A**.** The most frequently reported solicited systemic ARs included pyrexia (SCTV01E-2 vs. SCTV01E, 11.7% vs. 7.9%), fatigue (SCTV01E-2 vs. SCTV01E, 8.4% vs. 3.7%), and headache (SCTV01E-2 vs. SCTV01E, 7.5% vs. 4.2%), as shown in the Appendix A. 

In terms of severity, most solicited ARs were Grade 1 or 2, while six participants reported Grade 3-or-above pyrexia (SCTV01E-2 vs. SCTV01E, 5 [2.3%] vs. 1 [0.5%]); one (0.5%) participant in the SCTV01E-2 group reported Grade 3-or-above arthralgia.

### 3.7. Subgroup Analyses of Adverse Events

Stratified analysis by age revealed a numerically lower occurrence of treatment-related adverse events (TRAEs) in participants aged ≥60 years compared to those aged 18–59 years. Specifically, in the SCTV01E-2 group, 31.4% of participants aged ≥60 years experienced TRAEs, as opposed to 41.4% of participants aged 18–59 years. A similar pattern was observed in the SCTV01E group, with incidences of 32.9% and 33.8% among participants aged ≥60 years and 18–59 years, respectively. 

In the SCTV01E-2 group, two (2.3%) participants aged ≥60 years reported Grade 3-or-above TRAEs (one each reported pyrexia and arthralgia), and four (3.1%) participants among the young participants (18–59 years) reported Grade 3 pyrexia. In the SCTV01E group, only one (0.8%) participant aged 18–59 years reported Grade 3-or-above TRAEs (pyrexia). The occurrence of TRAEs in participants with pre-existing comorbidities was similar to that observed in the overall safety analysis population, with a 38.8% incidence in the SCTV01E-2 group and a 38.1% incidence in the SCTV01E group. 

## 4. Discussion

In this phase 2 trial, SCTV01E-2 demonstrated superiority over its progenitor vaccine SCTV01E in GMT and SRR of nAb against Omicron EG.5 sublineages, meeting the primary endpoint of this study. Both SCTV01E and SCTV01E-2 elicited a robust neutralizing response against the Omicron sublineages EG.5 and XBB.1 when administrated as a booster dose. The safety profile of SCTV01E-2 was found to be comparable to that of SCTV01E.

This study included a substantial high-risk demographic, with 40% of the participants being over 60 years of age and 42% having chronic comorbidities. The results indicated that SCTV01E-2 elicited nAb responses at levels comparable to those observed in the younger participants or those without underlying health conditions. Both SCTV01E-2 and SCTV01E were well tolerated across all age groups and medical histories, with the majority of AEs being mild and transient. Importantly, no vaccine-related SAEs, AESIs, or deaths were reported.

Multivalent vaccines represent an important strategy for the development of broad-spectrum vaccines, as each variant contributes unique neutralizing epitopes that expand the repertoire of neutralizing antibodies and the very frequent mutations occurring in multiple circulating variants likely present in future emerging variants. SCTV01E-2 and SCTV01E exemplify such multivalent COVID-19 vaccines. SCTV01E is derived from the bivalent vaccine SCTV01C, while SCTV01E-2, designed as an antigen-adapted vaccine based on SCTV01E, adjusts its antigen composition to align with the emerging strains, which include the Beta, BA.1, BQ.1.1, and XBB.1 variants. 

The selection of Beta variant as an antigen component was informed by its strong immune escape potential and its capabilities to induce cross-neutralizing activities against Omicron sublineages. The Beta variants possesses nine mutations in their spike protein, with N501Y, E484K, and K417N mutations promoting the immune escape and enhancing the binding affinity to human angiotensin-converting enzyme 2 (hACE2), which poses a challenge to COVID-19 vaccines designed based on the prototype variant [23,24]. A meta-analysis demonstrated a significant reduction in the effectiveness of prototype vaccines against the Beta variant compared to the Alpha and Delta variants [25]. Vaccines incorporating Beta variant, such as the Moderna mRNA-1273.211 (with a 1:1 mix of wild-type and Beta variants), SCTV01C (with a 1:1 mix of Alpha and Beta variants), and Vidprevtyn Beta (containing a spike protein of the Beta variant), exhibited robust cross-neutralization against multiple SARS-CoV-2 lineages. For instance, studies on mRNA-1273.211 revealed highly potent neutralizing antibody responses against the D614G, Beta, Delta, and Omicron variants one month post-booster [26]. Similarly, SCTV01C in mice has demonstrated broad-spectrum cross-neutralizing activities against 14 kinds of genetically distinct lineages of SARS-CoV-2 variants [27]. Additionally, Vidprevtyn Beta showed superiority against Omicron BA.1 and Omicron BA.4/5 variants one month after booster vaccination [28]. 

The level of nAbs is a key determinant and strong predictor of immune protection against symptomatic SARS-CoV-2 infection [29]. In this study, SCTV01E-2 demonstrates superior neutralization against Omicron EG.5, with significantly greater increases in GMT and SRR compared to SCTV01E. Considering that SCTV01E has already proven its efficacy in preventing symptomatic COVID-19 in a phase 3 study and has received authorization for use in China, it is anticipated that SCTV01E-2 will also demonstrate efficacy against SARS-CoV-2 infection, particularly for the circulating EG.5 and its sublineages.

The current rapid spread of the EG.5 variant, indicated by its reproductive (R) number, suggests a competitive advantage and implies that EG.5 and its sublineages are likely to continue dominating in the coming months. In this study, SCTV01E-2 induced significant nAb responses against both antigen-matched variant XBB.1 and antigen-mismatched variant EG.5, aligning with similar results from Moderna’s XBB.1.5-containing mRNA COVID-19 vaccine, which also elicited neutralizing responses against various XBB-lineage variants (XBB.1.5, XBB.1.6, and XBB.2.3.2) and recent variants (EG.5.1 and FL.1.5.1) [30,31]. These findings suggest that XBB-containing COVID-19 vaccines could induce nAb responses against subsequent XBB.1 lineages [30,31]. SCTV01E-2 is a viable vaccine option to prevent outbreaks of the possible EG.5 variant and its sublineages.

This report has a few limitations. First, it only presents the nAb response on Day 14 post-vaccination and the short-term safety profile. The study will continue to monitor participants for a minimum of 12 months to evaluate the duration of nAb and its long-term safety. Second, the percentage of participants with SARS-CoV-2 infection may be underestimated, given the difficulty of distinguishing between infection and vaccination through serological tests, especially in the context of the widespread use of inactivated COVID-19 vaccines [32,33]. Third, one of the study’s goals was to assess vaccine efficacy. However, at the time of the data cutoff, no cases of COVID-19 were identified in either group.

## 5. Conclusions

In summary, the formulation of SCTV01E-2 demonstrated advantages over its progenitor vaccine SCTV01E, particularly against the latest SARS-CoV-2 VOI, EG.5, with no safety concerns. As an antigen-adapted vaccine evolved from SCTV01E, SCTV01E-2 emerges as a promising candidate in the COVID-19 vaccine landscape, providing comprehensive protection against SARS-CoV-2 variants, particularly for older adults and individuals with chronic comorbidities.

## Figures and Tables

**Figure 1 vaccines-12-00175-f001:**
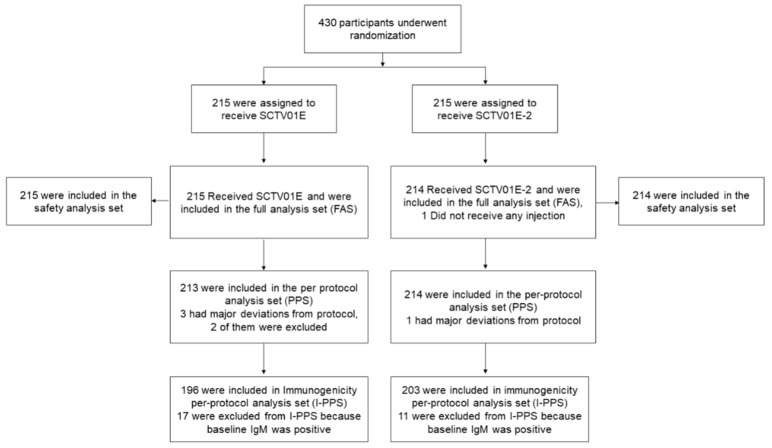
Trial profile. A total of 430 participants were recruited and randomized according to part A of this study; among them, the safety analysis population included 429 participants who had received the study vaccine. A total of 399 participants with a valid immunogenicity test result prior to and after the administration of study vaccines, a negative result of anti-spike receptor binding domain (RBD) IgM test at baseline, and no major protocol deviations were included in the immunogenicity per-protocol analysis population.

**Figure 2 vaccines-12-00175-f002:**
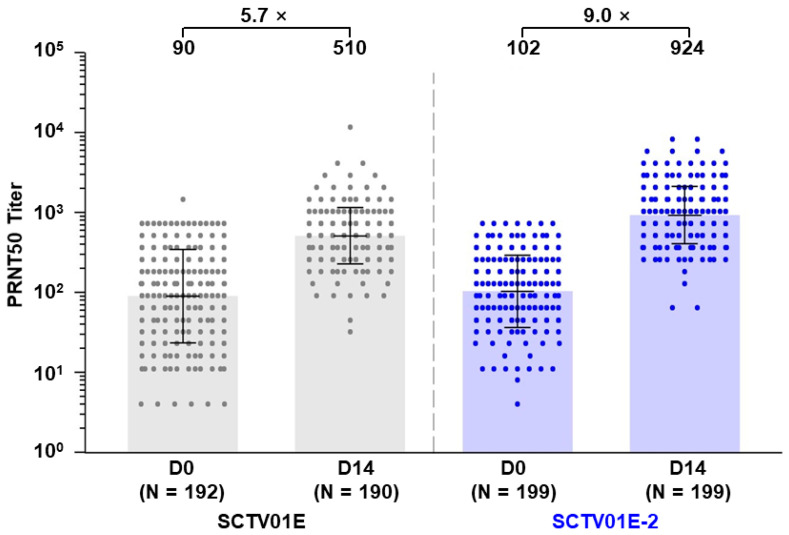
Neutralizing antibody titers against live SARS-CoV-2 Omicron variant EG.5 before and after vaccination. Titers of nAb were measured using a 50% plaque reduction neutralization test (PRNT50). Note: SCTV01E group (grey) and SCTV01E-2 group (blue). The center of the error bars represents the GMT. The bars represent the range from the GMT divided by the geometric SD factor to the GMT multiplied by the geometric SD factor. Dots represent the values of individual participants. Abbreviations: GMT, geometric mean titer; PRNT50, 50% plaque reduction neutralization test. SD, standard deviation.

**Figure 3 vaccines-12-00175-f003:**
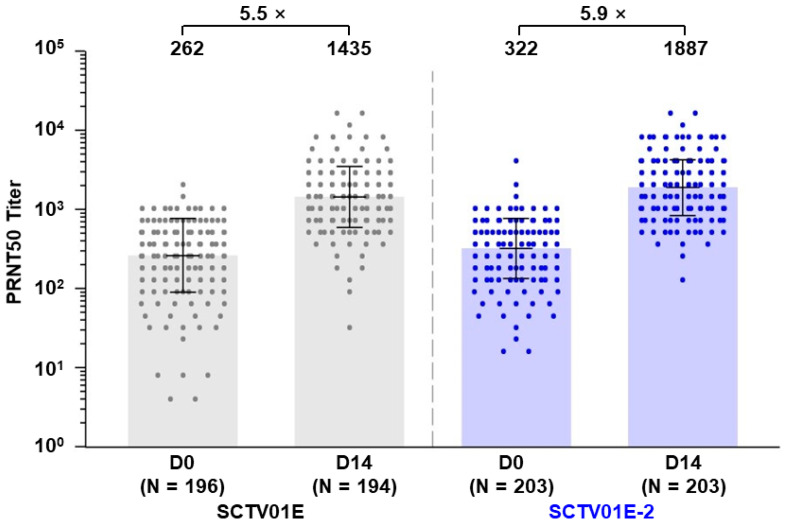
Neutralizing antibody titers against live SARS-CoV-2 Omicron variant XBB.1 before and after vaccination. Titers of nAb were measured using a 50% plaque reduction neutralization test (PRNT50). Note: SCTV01E group (grey) and SCTV01E-2 group (blue). The center of the error bars represents the GMT. The bars represent the range from the GMT divided by the geometric SD factor to the GMT multiplied by the geometric SD factor. Dots represent the values of individual participants. Abbreviations: GMT, geometric mean titer; PRNT50, 50% plaque reduction neutralization test. SD, standard deviation.

**Figure 4 vaccines-12-00175-f004:**
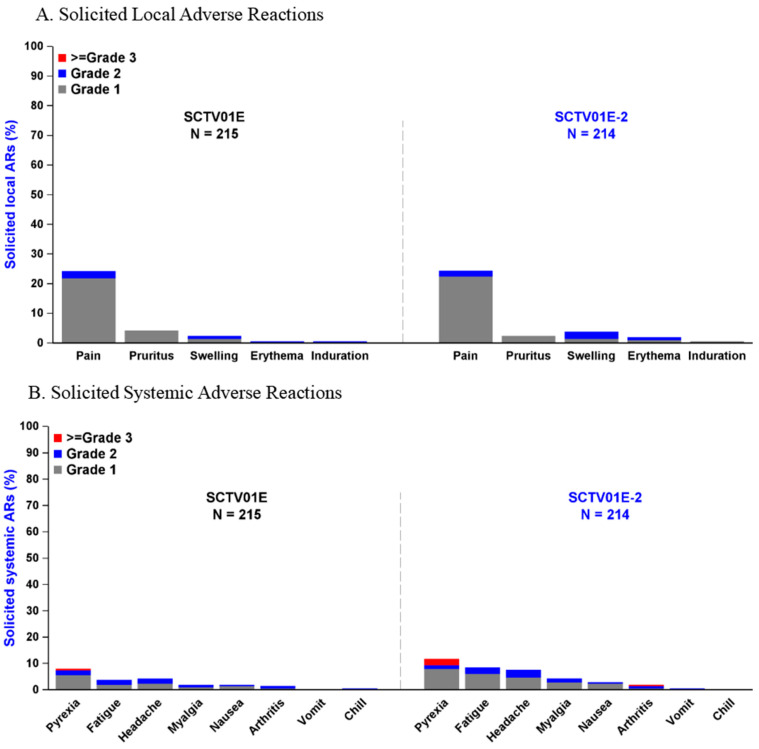
Incidence and grade of solicited local and systemic adverse reactions of the study vaccines. Safety analysis was conducted among participants who received the study vaccines. (**A**) The incidence and severity of solicited local adverse reactions within 7 days after the study vaccination. (**B**) The incidence and severity of solicited systemic adverse reactions within 7 days after the study vaccination. Adverse reactions were categorized as follows: Grade 1 for mild reactions, Grade 2 for moderate reactions, and Grade ≥3 for severe-and-above reactions.

**Table 1 vaccines-12-00175-t001:** Demographic characteristics of participants in the full analysis set.

	SCTV01E (N = 215) n (%)	SCTV01E-2 (N = 214) n (%)	Total (N = 429) n (%)
**Age (Years)**			
N	215	214	429
Mean (SD)	56.1 (12.1)	55.1 (11.5)	55.6 (11.8)
Median (Min, Max)	58.0 (20, 81)	58.0 (19, 83)	58.0 (19, 83)
**Age subgroups—randomization, n (%)**			
18–59 years	130 (60.5)	128 (59.8)	258 (60.1)
≥60 years	85 (39.5)	86 (40.2)	171 (39.9)
**Sex, n (%)**			
Male	101 (47.0)	98 (45.8)	199 (46.4)
Female	114 (53.0)	116 (54.2)	230 (53.6)
**Nation, n (%)**			
Han	214 (99.5)	214 (100.0)	428 (99.8)
Others	1 (0.5)	0	1 (0.2)
**BMI (kg/m^2^) ‡**			
N	215	214	429
Mean (SD)	26.2 (3.5)	26.5 (3.5)	26.4 (3.5)
Median (Min, Max)	26.0 (18.9, 38.6)	26.50 (16.4, 39.8)	26.20 (16.4, 39.8)
**History of SARS-CoV-2 infection, n (%)**			
Yes	57 (26.5)	56 (26.2)	113 (26.3)
No	158 (73.5)	158 (73.8)	316 (73.7)
**Previous vaccination/infection interval, n (%)**			
6–11 months	67 (31.2)	67 (31.3)	134 (31.2)
≥12 months	148 (68.8)	147 (68.7)	295 (68.8)
**IgM at baseline**			
Positive	17 (7.9)	11 (5.1)	28 (6.5)
Negative	198 (92.1)	203 (94.9)	401 (93.5)
**Booster dose of COVID-19 vaccine, n (%)**			
Yes	157 (73.0)	155 (72.4)	312 (72.7)
No	58 (27.0)	59 (27.6)	117 (27.3)
**Type of last received COVID-19 vaccine—randomization, n (%)**			
Inactive vaccine	98 (45.6)	103 (48.1)	201 (46.9)
Adenovirus vector vaccine	42 (19.5)	35 (16.4)	77 (17.9)
Recombinant protein vaccine	75 (34.9)	76 (35.5)	151 (35.2)
Other vaccines	0	0	0
**Pre-existing comorbidities, n (%)**			
Yes	84 (39.1)	98 (45.8)	182 (42.4)
No	131 (60.9)	116 (54.2)	247 (57.6)

‡ BMI, the body mass index. The body mass index is the weight in kilograms divided by the square of the height in meters.

## Data Availability

The authors also declare that the data supporting the findings of this study are available within the main manuscript or the Appendix A. Correspondence and requests for materials should be addressed to L.X. (lx@sinocelltech.com).

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
