# Peer review of "Immunogenicity of Tetravalent Protein Vaccine SCTV01E-2 against SARS-CoV-2 EG.5 Subvaraint: A Phase 2 Trial"

_vaccines, 2024, doi:10.3390/vaccines12020175_

Round 1

Reviewer 1 Report

Comments and Suggestions for Authors

This manuscript describes a promising new booster vaccine, SCTV01E-2, effective against emerging Omicron subvariants of SARS-CoV-2. Updating COVID-19 vaccines to combat continually evolving viral variants is crucial. This research supports the development of multivalent vaccines targeting multiple variants, a promising strategy for long-term COVID-19 control. Although, I wish the authors would explain in the discussion section their reasoning for including in the vaccine antigens of very old variants of the virus-long gone from circulation.

The study demonstrates convincingly an improved neutralizing antibody response compared to the parent vaccine. SCTV01E-2 elicited substantially higher neutralizing antibody levels against Omicron EG.5 than its predecessor, SCTV01E, indicating potential for enhanced effectiveness. Moreover, the new vaccine induced antibody responses against the XBB.1 variant, suggesting its potential for broader protection. Overall, the research provides compelling evidence for the potential of SCTV01E-2 as an effective and safe vaccine against Omicron EG.5 and possibly other SARS-CoV-2 variants. However, while the study is significant and offers convincing data, the clarity of the text and data presentation could be substantially improved.

In the present version of the text, there are many sloppy and unfortunate formulations. There are only three illustrations in the article and their titles are not signed correctly. The main conclusions of the article are not accurately stated. In the attached file, I have highlighted areas of the text that need significant work to improve clarity. In some places, I have given examples of how the wording could be improved. I would like to wish the authors to invest time and effort in the text and to elaborate it thoroughly. Good and reliable results should be accompanied by their clear presentation for the scientific community.

Comments on the Quality of English Language

There are almost no errors in grammar, punctuation or spelling in the text of the article. However, there are many problems in wording and clarity of presentation. It is the text that the authors need to seriously work on so that the high quality of their scientific work corresponds to the high quality of presentation.

Reviewer 2 Report

Comments and Suggestions for Authors

The article uses good vocabulary, is well written and includes all relevant data. However, shortcomings and ambiguities were found.

Notes to include:

1. How (by what test) was the statistical significance of differences between groups proven in the case of GMT administration? To determine the differences between SCTV01E and SCTV01E-2 groups, they must be confirmed. Throughout the work, if comparable/similar, higher or lower levels are claimed, it must be determined whether this is significant and with what probability (p).

2. Please present the endpoints of the study in a more understandable, parametric manner.

Line 171 – "(Error! Reference source not found.)." For this purpose, reference is made to the statistical parameters given below and Figure 1. There is no need to provide parameters adopted by other authors.

Line 180- "The immunogenicity per-protocol analysis population". Why per-protocol and not protocol?

Table 1 - headers in the table such as "Age subgroups-randomization, n (%)", "History of SARS-CoV-2 infection, n (%) " etc. cannot be in one column because they apply to all columns with data. I suggest moving them to column 1 and bolding them.

Line 187 - why per-protocol set and not protocol set?

Line 189 – please provide Table S1

Figure 2 I Figure 3 - If possible, please include the scatter whiskers on the GMT chart as Geo SD, the range from the GMT divided by the geometric SD factor to the GMT multiplied by the geometric SD factor will contain about two thirds of the values if the data are sampled from a lognormal distribution. A 95% CI seems inappropriate for such a spread of results (it is too low as if it were 5%).

Line 248 - please provide the observation period after vaccination, not the date and effectiveness during this period.

Comments on the Quality of English Language

Minor editing of English language required e.g. in Table S1 there is a nation, not a nationality.

Round 2

Reviewer 2 Report

Comments and Suggestions for Authors

The authors took into account all key comments and the article meets the requirements for publication in "Vaccine".

If possible, please correct the probability marked as "P" to "p" throughout the article, in accordance with the recommendations of statistical societies.